# High thermoelectric efficiency realized in SnSe crystals via structural modulation

Bingchao Qin [1,5], Dongyang Wang [2,5], Tao Hong [1,5], Yuping Wang [1], Dongrui Liu[1], Ziyuan Wang [3], Xiang Gao[4], Zhen-Hua Ge [3] & Li-Dong Zhao [1] ✉

Crystalline thermoelectrics have been developed to be potential candidates for power generation and electronic cooling, among which SnSe crystals are becoming the most representative. Herein, we realize high-performance SnSe crystals with promising efficiency through a structural modulation strategy. By alloying strontium at Sn sites, we modify the crystal structure and facilitate the multiband synglisis in p-type SnSe, favoring the optimization of interactive parameters $\mu$ and $m^*$. Resultantly, we obtain a significantly enhanced $PF$ ~85 $\mu$W cm$^{-1}$ K$^{-2}$, with an ultrahigh $ZT$ ~1.4 at 300 K and $ZT_{ave}$ ~2.0 among 300–673 K. Moreover, the excellent properties lead to single-leg device efficiency of ~8.9% under a temperature difference $\Delta T$ ~300 K, showing superiority among the current low- to mid-temperature thermoelectrics, with an enhanced cooling $\Delta T_{max}$ of ~50.4 K in the 7-pair thermoelectric device. Our study further advances p-type SnSe crystals for practical waste heat recovery and electronic cooling.

The carbon peaking and carbon neutrality goals have put forward higher, and more urgent requirements to improve the current energy efficiency as well as develop more green energy technologies[1–3]. Among them, thermoelectric (TE) technology realizes the direct conversion between heat and electricity, playing an increasingly significant role in the current global energy development strategy[4–8]. The keys to achieving TE conversion and its practical applications are to obtain stable and capable energy conversion efficiency in TE devices[7,9]. The device efficiency is largely determined by material property, defined as the dimensionless figure of merit $ZT$ with $ZT = (S^2\sigma)T/\kappa_{tot}$, in which $S$, $\sigma$, $T$, and $\kappa_{tot}$ refer to the Seebeck coefficient (thermopower), electrical conductivity, absolute temperature and total thermal conductivity (consisting of the electronic ($\kappa_{ele}$) and lattice ($\kappa_{lat}$) parts), respectively.

The above TE parameters possess strong and complex coupling interactions between each other, making it hard to decrease thermal transport while doing no harm to electrical properties (power factor, $PF = S^2\sigma$), and vice versa[6,10]. Even though, TE researchers have

established various strategies to realize the synergy and compromise between these interrelated parameters and enhance the final $ZT$, typically including carrier concentration tuning[11–13], carrier mobility optimization[14–17], band structure engineering[18–20], full-scale microstructure design[21–23], atomic-structure manipulation[24–26], and phonon-electron transports decoupling[27,28]. Traditionally, TE researches have focused on developing efficient transport properties for ingots and sintered/densified polycrystalline materials[21,22]. However, in recent years, crystalline TE materials have been continuously investigated in depth mainly due to their unique performance advantages[29–32].

Among them, tin selenide (SnSe) possesses unique transport features[33–37], being the most representative. Since the breakthrough discovery of the extraordinary high-temperature performance with $ZT$ ~2.6 ± 0.3[30], extensive achievements on the thermoelectric performance as well as the transport characteristics were realized[27,38–45]. Specifically, the complex electronic band structure and the momentum and energy multiband synglisis facilitate high performance in p-type SnSe[44,45], while 3D-charge/2D-phonon transports and

[1]School of Materials Science and Engineering, Beihang University, Beijing 100191, China. [2]Henan Key Laboratory of Diamond Optoelectronic Materials and Devices, Key Laboratory of Material Physics, Ministry of Education, School of Physics, Zhengzhou University, Zhengzhou 450052, China. [3]Faculty of Materials Science and Engineering, Kunming University of Science and Technology, Kunming 650093, China. [4]Center for High Pressure Science and Technology Advanced Research (HPSTAR), Beijing 100094, China. [5]These authors contributed equally: Bingchao Qin, Dongyang Wang, Tao Hong. ✉e-mail: zhaolidong@buaa.edu.cn

manipulated layered phonon-electron decoupling contribute to ultrahigh $ZT$ values in n-type SnSe[27,43]. These advancements have significantly promoted their practical utilizations in power generation as well as electronic cooling, while further performance improvement and device investigation are still required to achieve considerable conversion efficiencies[1].

The crystal structure can directly determine the electronic and phonon band structures, thus largely affecting the physical transport properties of bulk materials[46]. Previous studies have revealed that SnSe experiences the continuous phase transition, from a low-symmetric $Pnma$ to a high-symmetric $Cmcm$ phase, starting at ~600 K and completing the phase transition at ~800 K ($T_{pt}$)[43]. Moreover, it can be unraveled and summarized that a variety of performance optimization strategies that favor the thermoelectric properties of SnSe were essentially derived from the promoted phase transition by lowering $T_{pt}$ or improving crystal symmetry at low temperatures[27,40,44,47].

Herein, focusing on the crystal structure and based on the investigation on Pb-alloyed SnSe[44], we further introduced strontium (Sr) at Sn sites to modulate the crystal structure since SrSe possesses a NaCl-type high-symmetric cubic structure[48]. The Na-doped SnSe-9% Pb-x%Sr (x = 0.8, 1.2, 1.6, and 2.0) crystals were obtained with enhanced thermoelectric performance. The structural modulation by alloying Sr improves the crystal symmetry of low-temperature $Pnma$ phase to enhance carrier mobility $\mu$, promotes the multiband synglisis to decouple $\mu$ and effective mass $m^*$, which we further confirmed by using SR-XRD measurements and Rietveld refinements, electronic band structure calculations, and microstructure observations. Resultantly, we realized a significantly improved $\mu$ of ~326 cm$^2$ V$^{-1}$ s$^{-1}$, leading to the ultrahigh $PF$ of ~85 μW cm$^{-1}$ K$^{-2}$ with $ZT$ of ~1.4 at ambient temperature. The maximum average $ZT$ ($ZT_{ave}$) approaching 2.0 at 300–673 K was realized in the sample alloyed with 1.2% Sr, implying the great potential for low- to mid-temperature power generation. Moreover, we fabricated two single-leg thermoelectric devices using the obtained SnSe-9%Pb-1.2%Sr crystals with the plated Ni/Au bilayer being the contact

layer, achieving maximum conversion efficiencies of ~8.9 and ~8.8% at the temperature difference $\Delta T$ ~300 K according to the commercial Mini-PEM instrument. The stable and competent efficiencies of both devices further verified the performance optimization in p-type SnSe crystals through the structural modulation strategy. On this basis, we further fabricated a seven-pair thermoelectric device using p-type SnSe-9%Pb-1.2%Sr crystals and n-type commercial Bi$_2$Te$_{2.7}$Se$_{0.3}$, producing an enhanced cooling temperature difference $\Delta T_{max}$ of ~50.4 K at ambient temperature. The current results will undoubtedly be important progress in putting forward the application of SnSe crystalline thermoelectrics in low- to mid-temperature power generation as well as room temperature solid-state cooling. And the structural modulation strategy might be well utilized in many other low-dimensional systems.

## Results

### Performance simulation considering the phase transition

We investigated the thermoelectric transport properties with changing crystal structure. Using theoretical calculations in p-type SnSe due to the SR-XRD measurements[44], we obtained the electronic band structures of full-$Pnma$, full-$Cmcm$ SnSe, and several intermediate states between $Pnma$ and $Cmcm$ phases, based on which we estimated the typical electrical parameters including $\mu$, $\sigma$, $S$, and $PF$. We then plotted the parameter-distribution contour mappings in Fig. 1 with dynamic carrier concentration $n$ and crystal structure from $Pnma$ to $Cmcm$. Consistent with inherent understanding and cognition[46], the low-symmetric $Pnma$ phase possesses a more complex band structure, larger effective mass $m^*$, and thus larger $S$, while higher $\mu$ and $\sigma$ can be realized in the high-symmetric $Cmcm$ phase. Moreover, these competing parameters demonstrate opposite trends with dynamic $n$ and crystal structure, implying that possibly optimal $PF$ values can be achieved during the phase transition.

Specifically, according to the current simulations from Fig. 1d, the highest $PF$ values exceeding 100 μW cm$^{-1}$ K$^{-2}$ are expected, in the

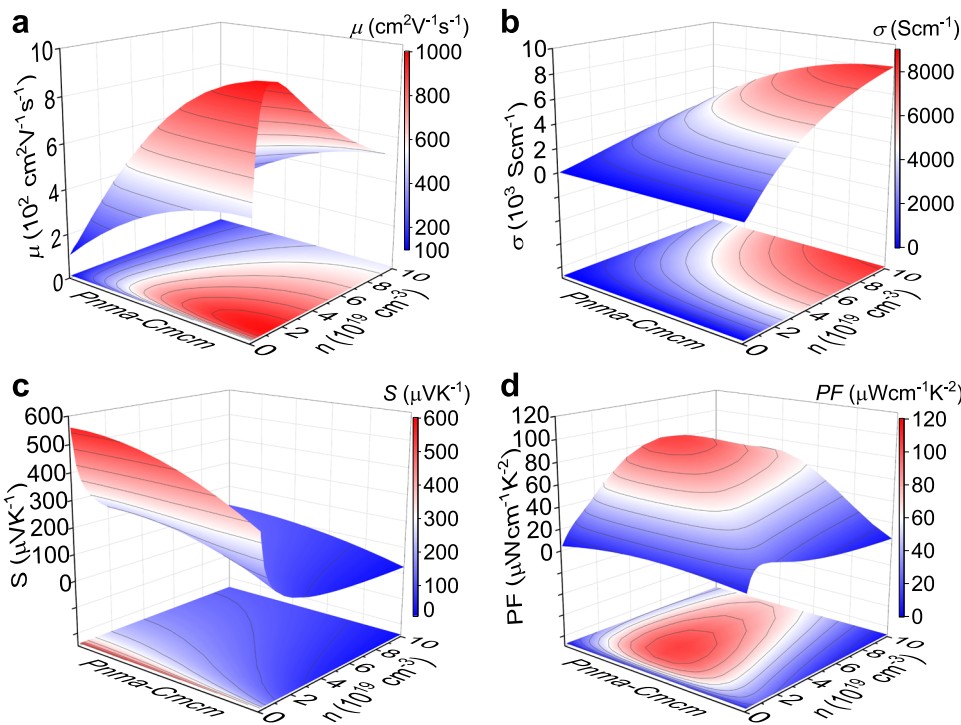

**Fig. 1 | Theoretical simulations on the electrical parameters with dynamic carrier concentration $n$ from $Pnma$ phase to $Cmcm$ phase at 300 K. a** Carrier mobility $\mu$. **b** Electrical conductivity $\sigma$. **c** Seebeck coefficient $S$. **d** Power factor $PF$.

condition with a crystal structure of ~60% *Pnma* or 40% *Cmcm* (assuming that full-*Pnma* is 100% *Pnma* or 0% *Cmcm*, and full-*Cmcm* is 100% *Cmcm* or 0% *Pnma*) and the $n$ ranging from ~3 to ~4 × 10$^{19}$ cm$^{-3}$. Notably, since these simulations were conducted on the basis of several assumptions, the obtained data of crystal structure or $n$ do not necessarily correspond to the maximum *PF* range. However, the conclusion is reasonable and trustworthy that the optimal electrical properties are expected in an intermediate state of the crystal structure between the *Pnma* and *Cmcm* phases.

## Electrical transport properties

In this study, we realized the structural modulation by alloying Sr in Pb-alloyed SnSe. The Pb-alloyed SnSe crystals were systematically investigated in our previous work[44]. Briefly speaking, in Pb-alloyed SnSe, we revealed the significantly promoted multiband synglisis including the momentum alignment of bands merging process from VBM (1 and 2) merging into VBM (1 + 2) and the energy alignment of bands aligning between VBM 3 and VBM (1 + 2). The momentum alignment from two bands (before merging) to one band (after merging) largely increased the carrier mobility, while the energy alignment of multiple bands aligning substantially strengthened effective mass. Therefore, the promoted multiband synglisis lead to ultrahigh power factor and *ZT* values near room temperature and over the whole temperature range.

Following the above simulations and on the basis of our former research on p-type SnSe with Pb alloying[44], we then introduced Sr at Sn sites to further tune the crystal structure. We used the X-ray back-reflection Laue measurement to determine the in-plane (*b-c* plane) orientations and then measured the thermoelectric properties all along the *b*-axis. Results showed that improved electrical performance was obtained (Fig. 2). $\sigma$ increases with Sr alloying and approaches the highest value of ~2127 S cm$^{-1}$ around room temperature (Fig. 2a), demonstrating ~20% enhancement compared with that of the Sr-free sample (~1770 S cm$^{-1}$)[44]. While Sr alloying improves $\sigma$, it has little effect on *S* (Fig. 2b) since $n$ almost remains constant, ranging from ~3.94 × 10$^{19}$ to ~4.19 × 10$^{19}$ cm$^{-3}$. Resultantly, higher power factors (*PF*s)

were realized, with the highest value exceeding 85 μW cm$^{-1}$ K$^{-2}$ in a 1.2% Sr-alloyed sample. We further estimated the role of Sr alloying by calculating the weighted mobility $\mu_W$, which largely reflects the optimal degree of the electrical performance in thermoelectrics[49,50]. As shown in Fig. 2d, $\mu_W$ shows similar trends with *PF* after Sr alloying. Considering that $\mu_W$ is generally regarded as the combination of $m^*$ and $\mu$[51], we plotted the comparisons on *S* and $\mu$ between experimental results and multiband simulations at 300 K shown in Fig. 2e, f. While the experimental *S* values showed well consistent with the simulated curve from the four-band model[44], $\mu$ presented significant enhancement with slightly increased $n$ after Sr alloying. The experimental and simulational analyses on the electrical transports implied that Sr alloying facilitates the electrical performance by drastically enhancing the $\mu$ without affecting $m^*$, leading to notably enlarged *PF*s at 300–773 K.

## Structural modulation by alloying Sr

To further unravel the roles of Sr alloying in p-type SnSe, we conducted a series of analysis on the phase structure, crystal structure, and electronic band structure after Sr alloying. For the sample SnSe-9%Pb-1.2% Sr, we conducted high-temperature SR-XRD measurements and refinements at 300–873 K, as depicted in Fig. 3, Supplementary Fig. 1, and Supplementary Tables 1, 2. No additional impurity phases were introduced into the SnSe lattice upon 1.2%Sr alloying. The diffraction peaks show slight shifts with rising temperature, resulting in dynamic lattice parameters in Fig. 3b. Compared with the Sr-free sample[44], the lattice parameters for *b*- and *c*- directions become closer in the temperature range before phase transition, indicating that Sr alloying further modulates the crystal structure after Pb alloying. Moreover, Sr alloying facilitates the phase transition in SnSe, by significantly lowering the temperature of completing phase transition ($T_{pt}$) from ~800 K in hole-doped SnSe, ~748 K in SnSe-9%Pb[44] to ~723 K in SnSe-9%Pb-1.2%Sr.

Sr alloying promotes the phase transition by not only lowering $T_{pt}$, but also by modifying the crystal structure details over a wide temperature range. Figure 4a schematically depicts crystal structure

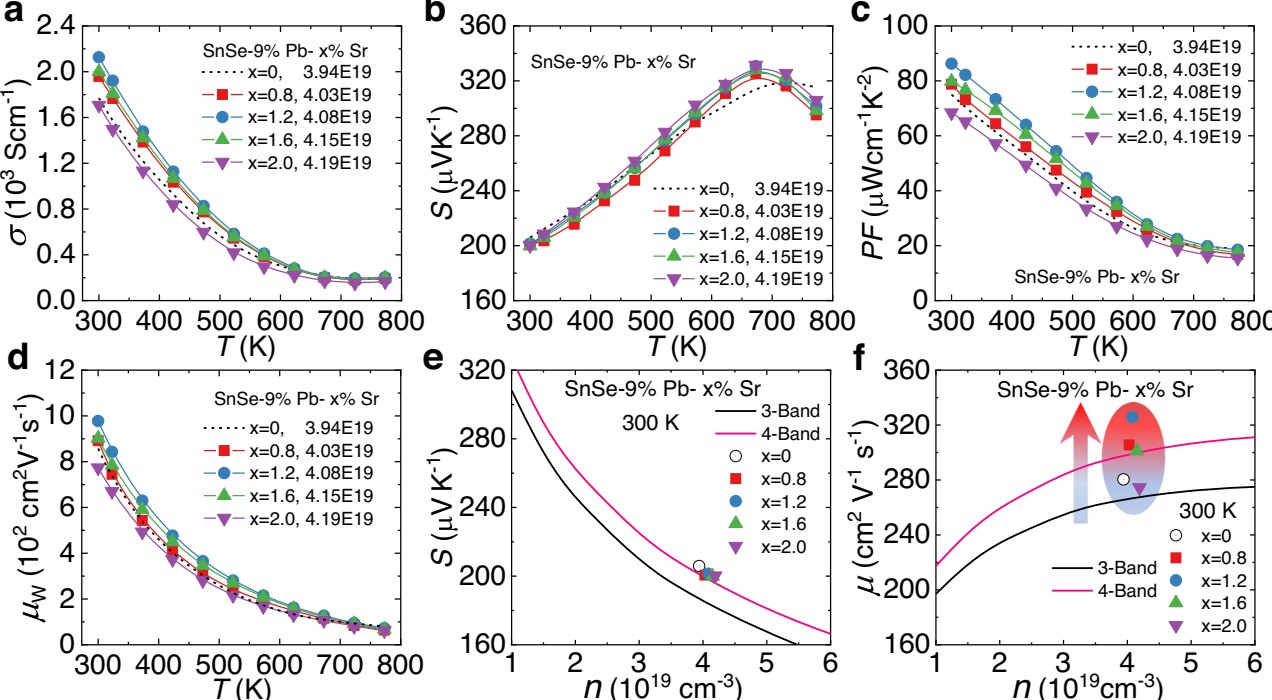

**Fig. 2 | Temperature-dependent electrical performance for SnSe-9%Pb-x%Sr crystals. a** Electrical conductivity $\sigma$. **b** Seebeck coefficient *S*. **c** Power factor *PF*. and **d** Weighted mobility $\mu_W$. Experimental data and multiband simulations with $n$ on the **e** Seebeck coefficient *S*. And **f** Carrier mobility $\mu$.

evolution along the in-plane direction, and the marked angle 1 turns to zero as crystal symmetry increases from *Pnma* to *Cmcm* phases. Based on the Rietveld refinements and obtained atomic positions in Supplementary Tables 1, 2, we obtained the angle 1 values for p-type SnSe, SnSe-9%Pb, and SnSe-9%Pb-1.2%Sr samples, as shown in Fig. 4b. The gradually decreasing angle values with temperature especially above ~600 K demonstrated the continuous phase transition process, while lowest values obtained in SnSe-9%Pb-1.2%Sr sample over a wide temperature range further confirmed the increased crystal symmetry after

Sr alloying. The modulated crystal structure symmetry contributes to the large enhancement of carrier mobility $\mu$ and electrical transport properties in Fig. 2.

In addition to the crystal structure modulation, Sr alloying also effectively adjusts the electronic band structure, especially the temperature-induced multiband evolution process of SnSe[44]. Based on the calculated band structures of SnSe-9%Pb-1.2%Sr sample at 300–873 K in Supplementary Fig. 2 according to the refinement results, we further distinguished the multiple valence bands, their

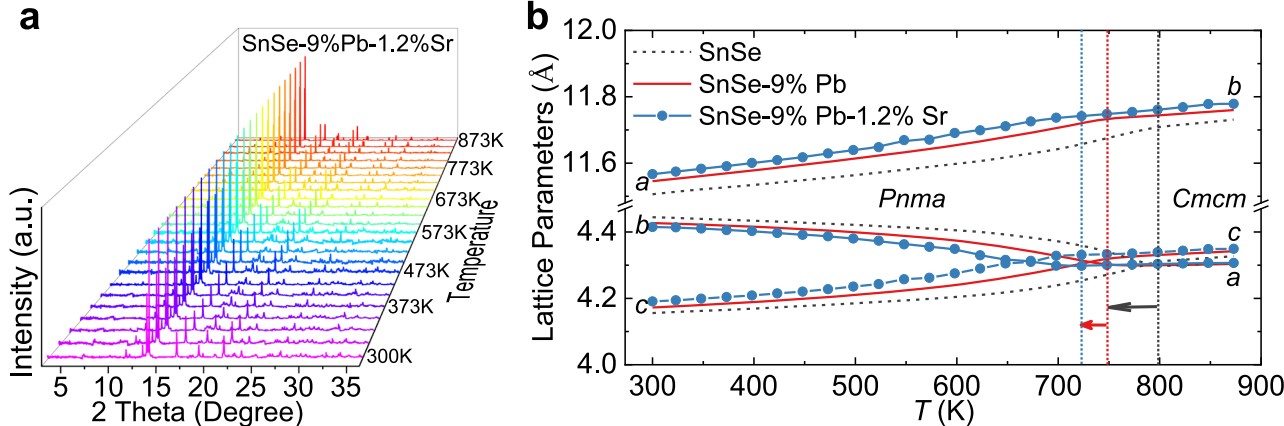

**Fig. 3 | SR-XRD measurements and Rietveld refinements of p-type SnSe-9%Pb −1.2%Sr crystal. a** Diffraction patterns from 300 to 873 K. **b** Temperature-dependent lattice parameters for hole-doped SnSe, SnSe-9%Pb, and SnSe-9%Pb −1.2%Sr. The arrows show that $T_{pt}$ declines from ~800 K in SnSe[44] to ~748 K in SnSe-9%Pb[44] and ~723 K for SnSe-9%Pb−1.2%Sr, indicating that Sr alloying further promotes the phase transition of SnSe.

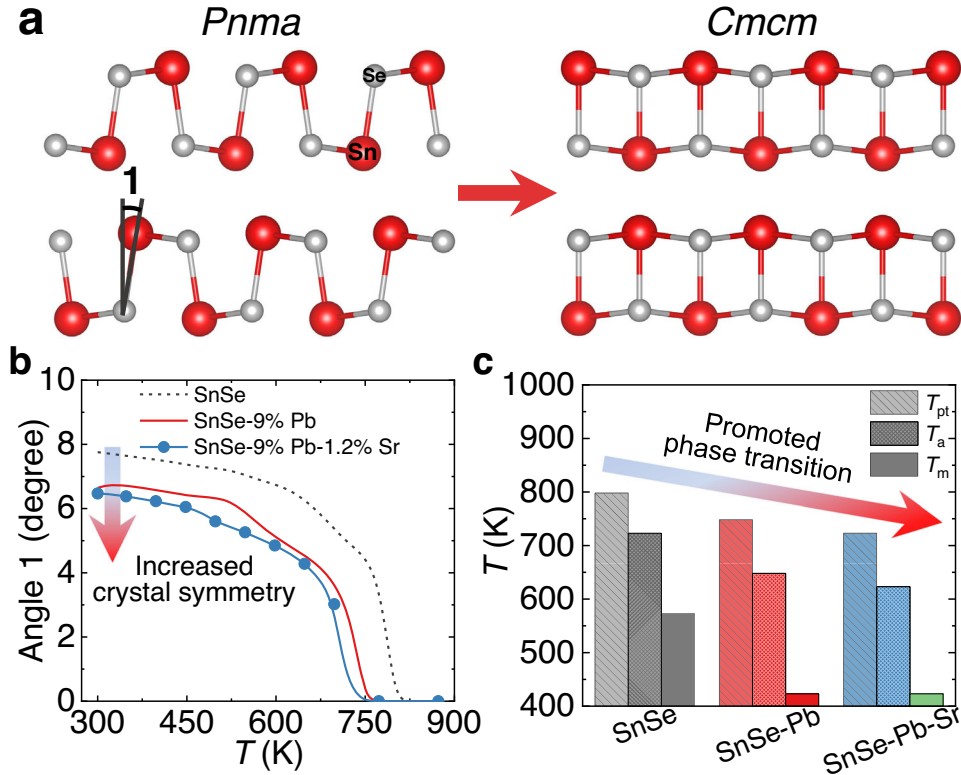

**Fig. 4 | Structural modulation by alloying Sr in SnSe. a** The crystal structure of *Pnma*- and *Cmcm*-SnSe, with the angle 1 marked, indicating the increased crystal symmetry. **b** The values of angle 1 in hole-doped SnSe[44], SnSe-9%Pb[44], and SnSe-9% Pb-1.2%Sr with temperature, showing that Sr alloying declines angle 1 and enhances the crystal symmetry over the whole temperature range. **c** The characteristic

temperatures for p-type SnSe[44], SnSe-9%Pb[44], and SnSe-9%Pb-1.2%Sr crystals, including the bands merging temperature ($T_m$) from VBM 1 and 2 to VBM (1 + 2), bands alignment temperature ($T_a$) between VBM 3 and VBM (1 + 2), and completing phase transition temperature ($T_{pt}$). The reduction of the characteristic temperatures implies that Sr alloying further promotes the multiband synglisis in p-type SnSe.

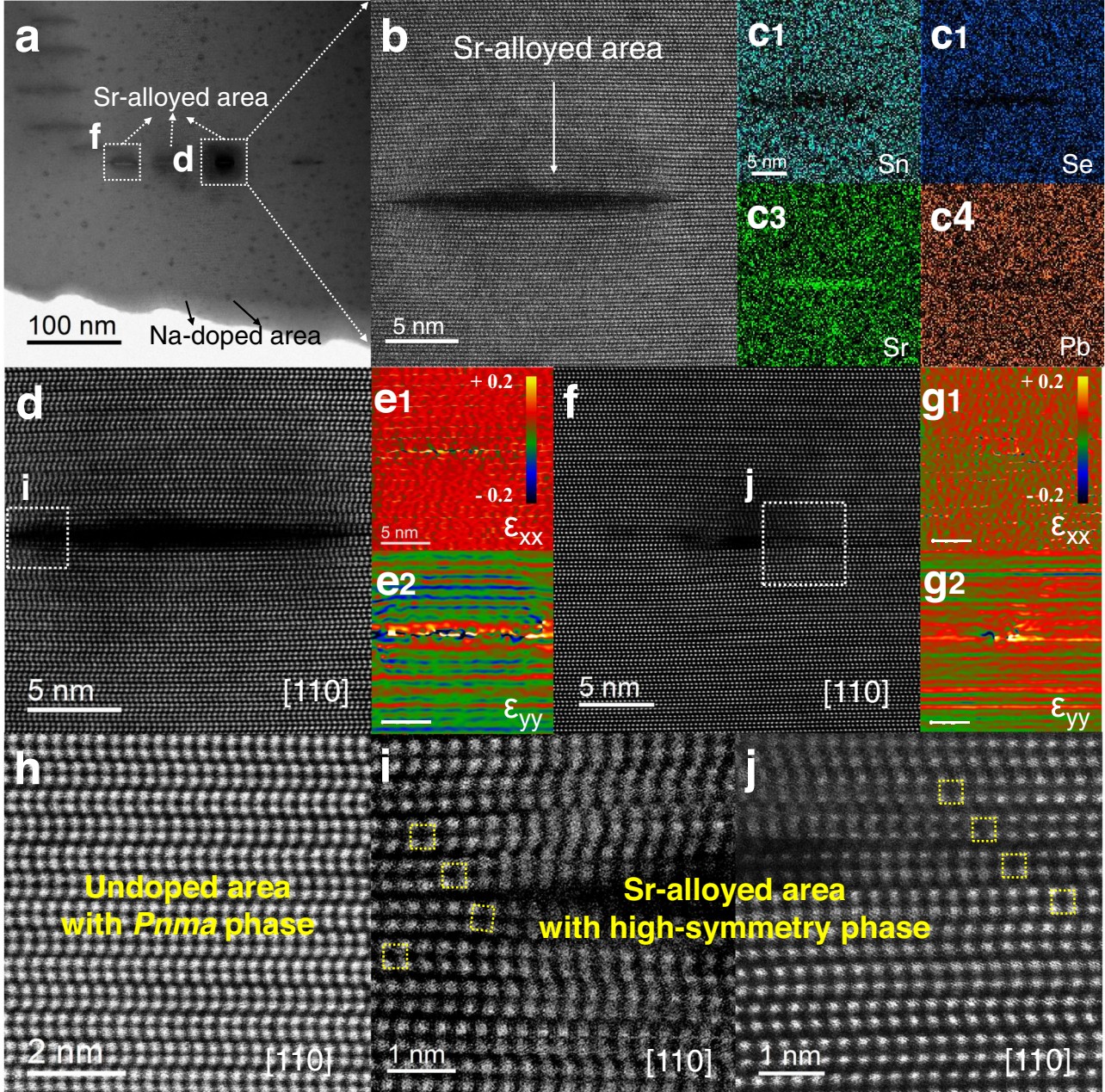

**Fig. 5 | Microstructure characterization on the p-type SnSe-9%Pb−1.2%Sr crystal. a** The ABF-STEM image exhibits the microstructure morphology of the sample. **b** The ADF-STEM image indicates the Sr-alloyed area. **c1**–**c4** Its corresponding elemental analysis. **d** The HAADF-STEM image of one Sr-alloyed area displays the expanded arrangement of atoms. **e1**–**e2** The corresponding GPA mappings along the horizontal (ε$_{xx}$) and vertical (ε$_{yy}$) axis. **f** Another Sr-alloyed region which reveals the contraction of atomic arrangements. **g1**–**g2** Its GPA maps. **h** The undoped area of zigzag structure with *Pnma* phase. **i, j** The enlarged atomic-resolved HAADF image demonstrates the high-symmetry phase around the Sr-alloyed area coming from **d** and **f**, respectively.

dynamic evolution with temperature, and energy differences between them in Supplementary Fig. 3. Our previous study has revealed the multiband synglisis including the bands merging process from VBM (1 and 2) to VBM (1 + 2), bands aligning process between VBM (1 + 2) and VBM 3[44], and Supplementary Fig. 3 further confirmed the processes. Moreover, Sr alloying further facilitates this dynamic evolution, by lowering the characteristic temperatures during the multiband synglisis. The bands merging temperature ($T_m$) from VBM (1 and 2) to VBM (1 + 2) was decreased from ~573 K in p-type SnSe to ~423 K in SnSe-9% Pb[44], and ~423 K in SnSe-9%Pb-1.2%Sr, while the bands aligning temperature ($T_a$) between VBM (1 + 2) and VBM 3 was estimated to be ~723, ~648, and ~623 K, respectively. Along with the declined phase transition temperature ($T_{pt}$), the reduction of the characteristic

temperatures based on Sr alloying depicted in Fig. 4c indicates the promotion of crystal structure and electronic band structure, generating significant synergy, especially between $\mu$ and $m^*$.

We further validated the structural modulation upon Sr alloying by regionally observing the structural details at the microscopic scale in Sr-substituted areas by scanning transmission electron microscopy (STEM) mode with Cs-corrected electron microscopy of JEM-ARM200F (JEOL). The annular bright-field (ABF) STEM picture for SnSe-9%Pb-1.2%Sr crystal was exhibited in Fig. 5a. There are two distinct types of contrasts in the morphology image, one of which appears like black spots deriving from the Na doping (marked by the black arrows) widely distributed in the sample. The other contrasts with the shape of a long stripe which are marked with white arrows. In

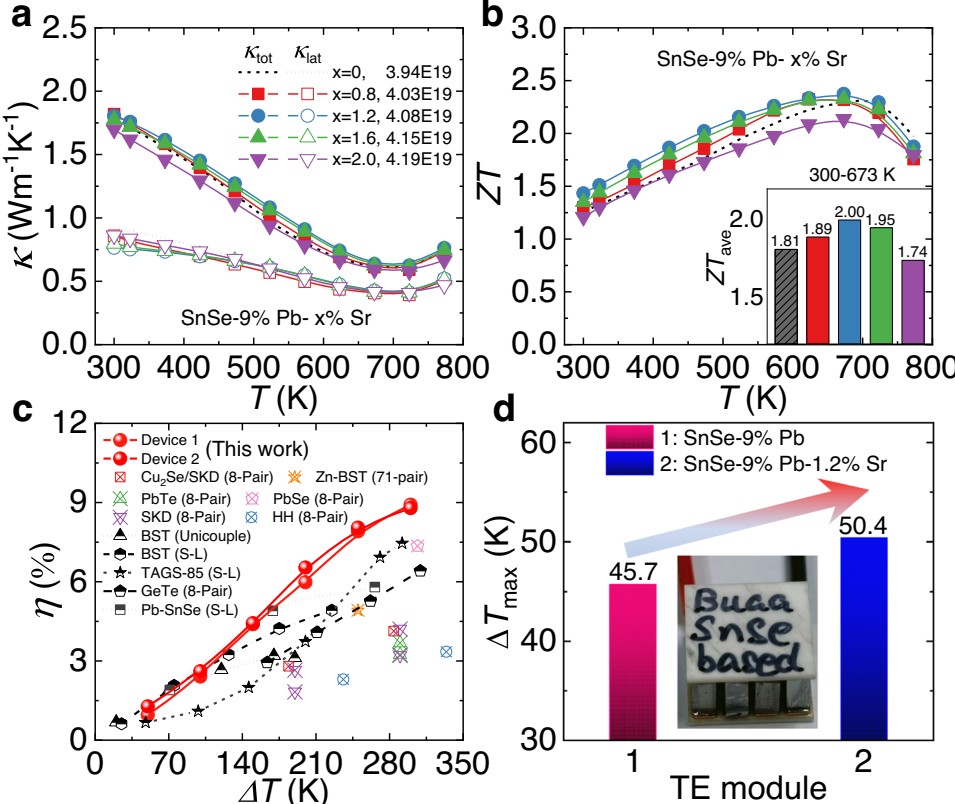

**Fig. 6 | Thermal transports, _ZT_, and device efficiencies for SnSe-9%Pb-x%Sr crystals. a** Total thermal conductivity $\kappa_{tot}$ and lattice thermal conductivity $\kappa_{lat}$. **b** _ZT_ and $ZT_{ave}$ at 300–673 K. **c** Comparisons on the experimental conversion efficiencies $\eta$ between the devices in this study (fabricated by using the SnSe-9%Pb−1.2%Sr crystals) and the reported results among comparable temperature differences $\Delta T$ (SKD skutterudites, BST $Bi_{2-x}Sb_xTe_3$, HH half-Heusler, TAGS-85 $(GeTe)_{85}(AgSbTe_2)_{15}$, S-L single-leg)[26,44,54–64]. **d** Maximum cooling temperature difference $\Delta T_{max}$ for the fabricated seven-pair thermoelectric devices using the SnSe-9%Pb–1.2%Sr crystals, and the $\Delta T_{max}$ for the SnSe-9%Pb crystals was plotted for comparison.

order to distinguish the constituents of the long stripe region, we obtained a magnified annular dark field (ADF) STEM picture and performed energy dispersive X-ray spectroscopy (EDS) analysis, which was shown in Fig. 5b, c1–c4. From the presented results, it can be concluded that this contrast region originates from Sr alloying.

Figure 5d depicts the high-angle annular dark field (HAADF) STEM picture of a Sr-alloyed region along the [110] direction, and it can be seen that the atomic arrangement close to the area is different from the arrangement far from this region. Furthermore, the corresponding geometric phase analysis (GPA) mappings shown in Fig. 5e1–e2 demonstrate strong negative stress along the vertical direction ($\varepsilon_{yy}$). There is another area of Sr alloying in Fig. 5f, which reveals the contraction of atomic arrangements. Similarly, its stress analysis in Fig. 5g1–g2 indicates that this area has strong positive stress along the vertical direction. Back to Fig. 5a, it can be observed that these two regions are in the same row, and the positive and negative stresses balance with each other at the macro level, thus maintaining the integrity of the crystal structure.

In order to investigate the variations in the surrounding regions of Sr alloying, further atomic level analysis was performed. Figure 5h shows the atomic arrangement of an undoped area with the _Pnma_ phase in the SnSe crystal along the [110] axis. As schematically shown in Supplementary Fig. 4b, c, there is a tiny angle between the Sn-Se bond and vertical axis in this [110] observation direction, corresponding with the angle in Fig. 4a, and this is the reason why this image shows a zigzag arrangement and maintains a long-range periodicity. Both Fig. 5i, j were the enlarged atomically resolved HAADF-STEM images around the Sr-alloyed region. Within the atomic layers marked by the yellow boxes, the arrangement of the atoms is close to the phase with higher

symmetry. This is the region that has been affected by the outward expansion of stress caused by Sr alloying. It can be concluded that the crystal structure symmetry of the neighboring areas can be improved due to the introduction of Sr, but no damage is created to its overall crystal structure, and SnSe crystals of high quality can still be obtained at the macroscopic level.

## Thermal transport properties, _ZT_, and device efficiencies

On the basis of the significantly facilitated electrical performance, Sr alloying slightly depress the thermal transport of SnSe. Due to the relevant parameters presented in Supplementary Fig. 5 and Supplementary Table 3, we evaluated the thermal properties depicted in Fig. 6a. Both the total ($\kappa_{tot}$) and lattice ($\kappa_{lat}$) thermal conductivities remain relatively low values due to the intrinsic strong bond anharmonicity[30,36]. Specifically, $\kappa_{tot}$ stays lower than 1.8 W m$^{-1}$ K$^{-1}$ at room temperature and gradually declines as the temperature rises until ~700 K, after which the upturn of $\kappa_{tot}$ and $\kappa_{lat}$ beyond 700 K originates from the completed phase transition. Moreover, $\kappa_{lat}$ shows a slight reduction with Sr alloying and approaches lower than ~0.8 W m$^{-1}$ K$^{-1}$. This can be arising from the strengthened phonon scattering from mass and strain-field fluctuations induced by Sr substituting Sn in the lattice[52,53], with minimum $\kappa_{lat}$ values approaching ~0.4 W m$^{-1}$ K$^{-1}$ around 700 K. Based on those synergically modulated electrical and thermal performance, we further estimated the quality factor, _B_, to comprehensively evaluate the optimized thermoelectric properties[49]. As shown in Supplementary Fig. 6, large enhancements in _B_ factors were detected over a wide temperature range, especially near 300 K, implying the significant adjustment between electrical and thermal transports based on Sr alloying.

Resultantly, we attained the optimal figure of merit $ZT$ values upon Sr alloying, as demonstrated in Fig. 6c. Obvious improvement on $ZT$, especially at low temperatures, can be realized, with a maximum $ZT$ of ~1.4 at 300 K in the sample SnSe-9%Pb-1.2%Sr, presenting a ~16% increase from Sr-free sample[44]. Moreover, peak $ZT$ values of ~2.4 were obtained, causing the excellent average $ZT$ ($ZT_{ave}$) ~2.0 among 300–673 K in the sample SnSe-9%Pb-1.2%Sr. High $ZT_{ave}$ across wide temperature ranges directly contributes to the energy conversion efficiency $\eta$. We then fabricated two single-leg thermoelectric devices using our SnSe-9%Pb-1.2%Sr crystals and estimated the power generation performance by using a commercial Mini-PEM instrument (Supplementary Fig. 7). The voltage ($U$), output power ($P$), and $\eta$ with dynamic external current ($I$) at various temperature differences ($\Delta T$, 50–300 K) with the cold-end temperature ($T_c$) fixed at ~295 K were simultaneously measured, as depicted in Supplementary Figs. 8, 9 for the two single-leg devices. As depicted in Fig. 6d, the maximum $\eta$ values of ~8.9 and ~8.8% were achieved at the $\Delta T$ of ~300 K, outstripping most of the currently reported high-performance thermoelectrics among comparable temperature differences, including those of the single-leg and multi-pair devices of BST ($Bi_{2-x}Sb_xTe_3$)[54–56], PbTe/PbSe[26,57,58], GeTe[59,60], etc[44,61–64]. We noted that the experimental efficiencies were still lower than theoretical values, which might originate from the relatively high contacting resistance and the large difference in the dimensions between samples and the heating stage during the measurement. Both of them will inevitably cause heat loss during the measuring process, which will thus lead to higher heat flow $Q_c$ and lower experimental $\eta$ values since $\eta = P/(P + Q_c)$. Moreover, considering the cross-sectional dimensions of the two single-leg devices ($0.32 \times 0.39\ cm^2$ for device 1 and $0.36 \times 0.42\ cm^2$ for device 2), we estimated the output power densities ($P_d$) for the devices based on our SnSe-9%Pb-1.2%Sr crystals. Supplementary Fig. 10 demonstrates that peak $P_d$ values exceeding $1.0\ W\ cm^{-2}$ were achieved at the $\Delta T$ of ~300 K in both devices, also demonstrating certain pre-eminence at this temperature range. Besides, the comparable power generation properties between the two single-leg devices elucidated their good stability and repeatability. Both the obtained ultrahigh conversion efficiency and power density have shown that the developed high-performance Sr-alloyed SnSe crystals possess great application potential for power generation among low- to mid-temperature ranges. Moreover, we have further fabricated a seven-pair thermoelectric device using p-type SnSe-9%Pb-1.2%Sr crystals and n-type commercial $Bi_2Te_{2.7}Se_{0.3}$, producing a maximum cooling temperature difference $\Delta T_{max}$ of ~50.4 K at ambient temperature, as shown in Fig. 6d and Supplementary Fig. 11, which further confirms the great potential for wide-bandgap SnSe crystals to be used in thermoelectric cooling.

## Discussion

In this study, we realized high thermoelectric performance in SnSe-Pb-Sr crystals by using the structural modulation strategy. On the basis of Pb alloying, Sr alloying plays multiple roles in enhancing electrical transport properties by promoting crystal structure modification and valence band evolution. For crystal structure, Sr alloying facilitates the phase transition, thus improves crystal symmetry and carrier mobility $\mu$, while for band structure, it facilitates the multiband synglisis by lowering the characteristic temperatures $T_m$, $T_a$, and $T_{pt}$. The crystal and band structures modulations favor the synergy between various parameters, especially carrier mobility and effective mass, leading to extraordinary $PF$ exceeding $85\ \mu W\ cm^{-1}\ K^{-2}$ around room temperature. The ultrahigh power factors directly determine the optimized $ZT$ values, with an exceptional $ZT_{ave}$ ~2.0 at 300–673 K and high $ZT$ ~1.4 at 300 K in SnSe. Resultantly, extraordinary power generation performance of the conversion efficiency approaching ~9% with output power density exceeding $1.0\ W\ cm^{-2}$ were obtained. Moreover, the fabricated seven-pair thermoelectric device showed an enhanced cooling $\Delta T_{max}$ of ~50.4 K. Our results significantly move forward p-type

SnSe crystals to be utilized in real applications of waste heat recovery, as well as electronic cooling and the strategy of structural modulation, which can be further implemented in various systems.

High-performance p-type SnSe crystals and promising device efficiencies based on them were successfully realized by utilizing a structural modulation strategy. The crystal structure in solids directly determines the electronic and phonon band structures, which further have significant impacts on the physical transports, typically including thermoelectric performance. In recent years, focusing on modifying the crystal structure, especially its symmetry, has been well implemented in many low-dimensional thermoelectric materials to synergistically optimize their performance. For p-type SnSe crystals, the current materials' properties have shown certain advantages as low- to mid-temperature power generation candidates and even thermoelectric coolers. The next and more important step is to investigate and develop high-efficiency devices. To achieve this goal, focusing on the interfacial and geometrical design of the modules and devices will be essential.

## Methods

### Sample synthesis and crystal growth

In this study, high-quality $Sn_{0.895-x\%}Na_{0.015}Pb_{0.09}Sr_{x\%}Se$ (x = 0.8, 1.2, 1.6, and 2.0. And in this paper, these samples are abbreviated as SnSe-9%Pb-x%Sr crystals). The raw materials were accurately weighed and placed in quartz tubes, vacuumed, and sealed. Place the tubes into outer quartz tubes, vacuum, and seal again to protect the samples. Quartz tubes containing raw materials were placed in a vertical temperature-gradient furnace for crystal growth. The temperature program at the sample end of the furnace was set as follows: heat up to 1223 K within 15 h and hold for 15 h, then slowly cool down to 1073 K at a cooling rate of 1 K h$^{-1}$, and then cool with the furnace. Finally, crystal samples with a diameter of ~15 mm and a length of ~35 mm were obtained.

### X-ray back-reflection Laue

After obtaining the crystal samples, we used X-ray back-reflection Laue measurement to determine the specific orientations within the cleavage plane (400) of SnSe. During the test, the cleavage plane of the sample was facing the X-ray source, and the $b/c$ direction within the crystal plane can be determined by collecting diffraction spot information. All samples were cut in the $b$-axis direction for subsequent thermoelectric performance measurements.

### High-temperature synchrotron radiation X-ray diffraction (SR-XRD)

The p-type SnSe-9%Pb-1.2%Sr crystal sample was carefully grounded, sieved, placed in a quartz capillary in a glove box, and sealed. Synchrotron radiation measurements were carried out at the BL14B1 line station of the Shanghai Synchrotron Radiation Facility (SSRF), and high-energy X-ray diffraction data from 300 to 873 K were then obtained. Refinement of the diffraction data yields detailed information such as lattice parameters and atomic positions, as shown in Supplementary Fig. 1 and Supplementary Tables 1, 2, which were used for further analysis.

### Electrical and thermal transport properties

All crystalline samples were cut along the in-plane $b$-axis direction for measurement of thermoelectric performance parameters. Among them, using the Cryoall CTA and Ulvac Riko ZEM-3 instruments, based on the four-probe method, the electrical conductivity, and Seebeck coefficient were tested simultaneously on the long-strip sample with a size of ~$4 \times 4 \times 8\ mm^3$ under a helium atmosphere; Using the Netzsch LFA457 equipment, based on the laser flash method, the thermal diffusivity ($D$) of the disc-shaped sample with a diameter of about 6 mm and a thickness of 1–2 mm was tested under a flowing nitrogen

atmosphere. The thermal conductivity of the sample is calculated by using $\kappa = D \cdot \rho \cdot C_p$, where the density $\rho$ was tested by Archimedes method and reviewed by the gas pycnometer (Micromeritics AccuPyc II 1340) measurement, and the specific heat $C_p$ was calculated using the Debye model. Combining the electrical conductivity, Seebeck coefficient, and thermal conductivity obtained from the measurements, the $ZT$ values of the samples can finally be calculated. Taking into account all possible errors during these measurements, the resulting $ZT$ values have an uncertainty of less than 20%.

## Hall measurement

Samples with an in-plane diameter of ~8 mm and a thickness of ~0.8 mm were used for Hall measurement and carrier analysis. In a reversible magnetic field with a maximum magnetic field strength of ~1.5 T, the room temperature Hall coefficients ($R_H$) of the samples were tested by the Van der Pauw method using the commercial equipment (Lake Shore 8400 Series, Model 8404, USA), and the carrier concentration ($n_H$) was obtained by $n_H = 1/(e \cdot R_H)$. Combined with the electrical conductivity ($\sigma$) obtained by the above measurement, the carrier mobility ($\mu_H$) can be calculated by using $\mu_H = \sigma \cdot R_H$.

## Single-leg thermoelectric device fabrication and power generation performance measurement

The Mini-PEM from Advance Riko company was used to examine the power generation performance of two single-leg devices based on the SnSe-9%Pb-1.2%Sr crystals (Supplementary Fig. 7). In order to increase the roughness of the sample surface, the upper and lower surfaces of the samples with dimensions of ~4 × 4 × 8 mm³ were first processed by sanding the samples with 4000 grit sandpaper and etching the samples with an acidic solution. The treated samples were then electrochemically plated with a metallization Ni/Au bilayer. In order to further minimize the contact resistance at the interface, silver pastes were lastly put as a connecting layer between the Cu sheet and the Ni layer. The prepared single-leg thermoelectric device was set up on the testing equipment, and silicone grease was applied to the Cu sheet at the high-temperature end to improve thermal conductivity. The test temperature differences were set as 50, 100, 150, 200, 250, and 300 K, and the cold-end temperature was set as ~300 K. The measurement results for the two devices were provided in terms of output power, output voltage, and conversion efficiency (Supplementary Figs. 8, 9).

## Fabrication and cooling temperature difference testing of thermoelectric cooling devices

Using high-performance SnSe-9%Pb-1.2%Sr crystals for the p-type legs and commercially available $Bi_2Te_{2.7}Se_{0.3}$ as the n-type legs, a thermoelectric cooling device was constructed to estimate the cooling performance. The ends of the thermoelectric materials were acid washed and ground with 4000 grit sandpaper before being electroplated with a 4-µm-thick Ni diffusion barrier layer, and then coated with a 1-µm-thick Au film. The treated samples were divided into 2 × 2 × 4 mm³ pellets, which were then joined together using Sn-Bi solder to form two wires that were then connected to copper electrodes. Finally, we prepared a seven-pair thermoelectric cooling device with dimensions of 10 × 10 × 5.8 mm³. We used Z-Meters (a commercial thermoelectric cooler test equipment, RMT Ltd., Russia) to allow visual testing of the cooling performance parameters. The obtained result of internal resistance and $\Delta T_{max}$ of the thermoelectric device were shown in Supplementary Fig. 11.

## Microstructure characterizations

The transmission electron microscopy (TEM) specimen was prepared by traditional methods, including mechanical cutting, polishing, and then ion milled with Gatan PIPS II at 2.5 kV till hole formation and then cleaned using 1 kV and 100 V. The obtained ABF-STEM image, ADF-STEM image and the HAADF-STEM image were operated at ARM200F(C)-ONE ARM using the annular bright-field (ABF) detector and ADF detector.

## Data availability

The authors declare that the data supporting the findings of this study are available on reasonable request.

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

## Acknowledgements

This work was financially supported by the National Key Research and Development Program of China (2018YFA0702100), the National Natural Science Foundation of China (52250090, 51772012, 51571007, and 12204156), the Beijing Natural Science Foundation (JQ18004), the 111 Project (B17002), and the National Science Fund for Distinguished Young Scholars (51925101). L.-D.Z. acknowledges the support from the high-performance computing (HPC) resources at Beihang University, BL14B1 at Shanghai synchrotron radiation facility (SSRF) for the SR-XRD mea-surements, and Center for High-Pressure Science and Technology Advanced Research (HPSTAR) for STEM measurements. B.Q. thanks for the support from the Academic Excellence Foundation of BUAA for PhD Students.

## Author contributions

L.-D.Z. and B.Q. conceived the idea, designed the experiments, and supervised the research. B.Q. performed the sample synthesis, struc-tural characterization, and thermoelectric transport property measure-ments. D.W. carried out the theoretical calculations. X.G. and T.H. performed microstructure characterization of the samples. Z.W. and Z.-H.G. prepared the single-leg devices and carried out power generation tests. B.Q., Y.W., and D.L. carried out the high-temperature synchrotron radiation X-ray diffraction (SR-XRD) measurements and analyzed the data. B.Q. and D.L. fabricated the seven-pair thermoelectric device and measured the cooling performance. B.Q. and L.-D.Z. wrote the

manuscript with contributions from other authors. All authors analyzed the results and commented on the manuscript.

## Competing interests

The authors declare no competing interests.
