## [Peer Review File · Nature Communications]

High thermoelectric efficiency realized in SnSe crystals via structural modulationREVIEWER COMMENTS

Reviewer #1 (Remarks to the Author):

This study reports the improved thermoelectric performance of p-type SnSe by manipulating crystal lattice with Pb/Sr doping. The structural modulation contributes to the enhanced electrical performance, leading to a maximum power factor of $\sim 85 \mu\text{W cm}^{-1} \text{K}^{-2}$ and ZT of ~ 1.4 at 300 K. The high-efficiency, up to 8.9%, thermoelectric devices based on the current materials is further constructed with the newly contacting Ni/Au bilayer. This study demonstrated an important progress in advancing the practical application of SnSe crystals for waste heat recovery. Therefore, I recommend this manuscript for publication in Nature Communications after some minor revisions listed below.

(1) In Figure 4d, the authors announced that the bands merging temperature T_m was decreased from ~ 573 K in hole-doped SnSe, to ~ 448 K in SnSe-9%Pb, and ~ 423 K in SnSe-9%Pb-1.2%Sr. However, the reported T_m for SnSe-9%Pb was ~ 423 K in ref 44. The authors should check it.

(2) Since the phase transition temperature has been lowered to ~ 723 K in Sr-alloyed SnSe in this study, it seems to be meaningless to calculate the average ZT from 300 K to 773 K. The large temperature difference across phase transition will undoubtedly cause the failures of the device.

(3) The reported efficiency of thermoelectric device reaches 8.9%. This value is certainly high, but it is much less than the theoretical predication calculated with the averaged ZT of 2.0. The authors should give some comments on this discrepancy.

(4) The authors should list the densities of the crystalline samples in this work. Those data are important for the evaluation of crystal quality.

Reviewer #2 (Remarks to the Author):

In this study, Qin et al. reported significantly enhanced thermoelectric performance in SnSe crystals by using the structural modulation strategy. They also obtained extraordinary thermoelectric conversion efficiencies in SnSe-based devices, further developing SnSe crystals as great and potential candidates, especially for low-temperature power generation. By alloying Sr at Sn sites, both the crystal structure and electronic band structure were facilitated, leading to the simultaneous enhancement of carrier mobility and effective mass. Resultantly, they obtained ultrahigh power factor PF of $\sim 85 \mu\text{W cm}^{-1} \text{K}^{-2}$ and ZT of ~ 1.4 at 300 K in the sample SnSe-9%Pb-1.2%Sr. On this basis, they fabricated thermoelectric devices to re-confirm the superior materials' performance, attaining rather high and stable conversion efficiencies approaching 9% at the temperature difference of only ~ 300 K. Both the materials' performance and the devices' efficiencies reported in this study showed obvious superior among the current low- to mid-temperature thermoelectrics. Moreover, the structural modulation strategy was well interpreted and verified by experimental observations as well as theoretical calculations. Therefore, I recommend this article be accepted for publication in Nature Communications. And before the final publication, there are some minor revisions to be resolved.

1. I notice that this work can be regarded as a continuation of the paper published in 2021 (Science 2021, 373 (6554), 556-561), and the room-temperature ZT was further increased from ~ 1.2 to ~ 1.4 . While the power generation efficiency was largely enhanced, how about the thermoelectric cooling performance? Higher cooling temperature difference was also

expected in this work.

2. The performance simulations from full-Pnma phase to full-Cmcm phase in Figure 1 are quite interesting. It is easy to understand that the two end curves in the 3D figures were obtained using the calculated electronic band structure of Pnma phase and Cmcm phase due to SR-XRD results. However, we can't obtain the 3D figures only from two end curves. How to obtain the simulated performance curves of the intermediate states between full-Pnma phase and full-Cmcm phase?
3. The bands merging temperature T_m for SnSe-9%Pb was announced to be ~ 423 K, not ~ 448 K. Please check it.
4. The multiband simulation curves of 3-band and 4-band models were plotted in Figure 2e-f. It is required to provide the simulation details in the Supplementary Information.
5. In the first paragraph of the Introduction, κ_{lat} refers to the "lattice" part of κ_{tot} , not the "thermal" part.

Reviewer #3 (Remarks to the Author):

Tin-selenide-based materials have been considered as one of the most promising thermoelectric materials and they have been widely investigated. In this manuscript, Zhao et al. reported a structural modulation strategy to improve the thermoelectric performances of p-type SnSe crystals. By alloying strontium at Sn sites, the crystal symmetry of low-temperature Pnma phase was modified, and the low- to mid-temperature thermoelectric performance were largely enhanced. Resultantly, an ultrahigh $ZT \sim 1.4$ at 300 K and $ZT_{ave} > 2.0$ among 300 – 773 K are obtained. The experimental and theoretical works in this manuscript are well performed, and the manuscript is well written. However, there are few points need to be addressed and clarified. I suggest the authors consider the follow comments before the manuscript can be accepted for publication.

1. SnSe possesses a layered structure and anisotropic thermoelectric properties, therefore, the thermal and electrical properties of the present sample should be measured in the same direction. The authors are suggested to provide the details on the measurement direction specifically in the manuscript.

2. As reported in the manuscript, the Na-doped SnSe-9%Pb-x%Sr ($x = 0.8, 1.2, 1.6, \text{ and } 2.0$) samples were prepared in this work. A high $ZT \sim 1.2$ at 300 K is obtained, which is much higher than that of the reported Pb-doped SnSe and the Bi_{0.5}Sb_{1.5}Te₃ materials. However, the role of Pb alloying is not discussed in the manuscript. The authors should address this issue and make the comparison with reported Pb-doped SnSe sample.

3. As shown in Fig. 5, both the Pnma phase and the high-symmetry phase are observed in the as-prepared SnSe-9%Pb-1.2%Sr sample. Thus, the two phases should be considered during Rietveld refinements of p-type SnSe-9%Pb-1.2%Sr sample. Moreover, the refinements results for each sample should be provided in form of table in Supplementary Information.

4. In Fig. 5h, Fig. 5i and Fig. 5j, the black backgrounds of text box in the middle of the figures affected the display of pictures.

5. In addition, it is suggested to polish the English of the whole manuscript.

Authors Response

Reviewer #1 (Remarks to the Author)

General Comment: This study reports the improved thermoelectric performance of p-type SnSe by manipulating crystal lattice with Pb/Sr doping. The structural modulation contributes to the enhanced electrical performance, leading to a maximum power factor of $\sim 85 \mu\text{W cm}^{-1} \text{K}^{-2}$ and ZT of ~ 1.4 at 300 K. The high-efficiency, up to 8.9%, thermoelectric devices based on the current materials is further constructed with the newly contacting Ni/Au bilayer. This study demonstrated an important progress in advancing the practical application of SnSe crystals for waste heat recovery. Therefore, I recommend this manuscript for publication in *Nature Communications* after some minor revisions listed below.

Response: We thank Reviewer 1 for the positive comments and recognition on our study. We also appreciate those critical suggestions which help further improve the scientific quality of our manuscript.

Comment 1: In **Figure 4d**, the authors announced that the bands merging temperature T_m was decreased from ~ 573 K in hole-doped SnSe, to ~ 448 K in SnSe-9%Pb, and ~ 423 K in SnSe-9%Pb-1.2%Sr. However, the reported T_m for SnSe-9%Pb was ~ 423 K in ref 44. The authors should check it.

Response: Thanks for pointing out this question and we are sorry for our carelessness. We have fixed the relevant discussion and **Fig. 4** in the Revised Manuscript.

Revision:

Fig. 4c. The characteristic temperatures for p-type SnSe, SnSe-9%Pb, and SnSe-9%Pb-1.2%Sr crystals.

Comment 2: Since the phase transition temperature has been lowered to ~ 723 K in Sr-

alloyed SnSe in this study, it seems to be meaningless to calculate the average ZT from 300 K to 773 K. The large temperature difference across phase transition will undoubtedly cause the failures of the device.

Response: Thanks for your suggestion. We have calculated the ZT_{ave} at 300-673 K and plotted it in **Figure 6** in the Revised Manuscript.

Revision:

Fig. 6b. ZT and ZT_{ave} at 300-673 K for p-type SnSe-9%Pb-x%Sr crystals.

Comment 3: The reported efficiency of thermoelectric device reaches 8.9%. This value is certainly high, but it is much less than the theoretical predication calculated with the averaged ZT of 2.0. The authors should give some comments on this discrepancy.

Response: Thanks for your comments. The discrepancy between the obtained average ZT of ~ 2.0 and the measured efficiency of 8.9% originates from several aspects. Firstly, the average ZT of ~ 2.0 was realized in the temperature of 300 – 773 K, while the efficiency of $\sim 8.9\%$ was obtained at a temperature difference of ~ 300 K over the temperature range of 295 – 595 K. The ZT_{ave} at 295 – 595 K was estimated to be ~ 1.85 , producing a theoretical conversion efficiency η of $\sim 15\%$. Secondly, the discrepancy between the experimental η of $\sim 8.9\%$ and theoretical η of $\sim 15\%$ mainly comes from the high contacting resistance between the crystalline samples and the Cu metal sheets. The contacting layers with high electrical resistance causes extra heat absorption and heat loss between the heater and the sample, which largely enhances the heatflow Q_c and thus decreases the measured η since η is calculated by $\eta = P/(P+Q_c)$. Moreover, during the measurement, the dimensions of the heating stage is $\sim 10 \times 10$ mm², while the upper and lower cross-sectional dimensions of the samples are only $\sim 4 \times 10$ mm². The difference on the dimensions will also cause inevitable heat loss during the heating process, which will also lead to higher Q_c and lower experimental η values.

Comment 4: The authors should list the densities of the crystalline samples in this work. Those data are important for the evaluation of crystal quality.

Response: Thanks for your suggestion. We have provided the sample densities in the Revised Supplementary Information.

Revision:

Supplementary Table 3. Sample densities for SnSe-9%Pb-x%Sr crystals.

Sample (x)	Density (g cm ⁻³)
0	6.01
0.8	5.95
1.2	5.98
1.6	6.05
2.0	5.97

Reviewer #2 (Remarks to the Author)

General Comment: In this study, Qin et al. reported significantly enhanced thermoelectric performance in SnSe crystals by using the structural modulation strategy. They also obtained extraordinary thermoelectric conversion efficiencies in SnSe-based devices, further developing SnSe crystals as great and potential candidates, especially for low-temperature power generation. By alloying Sr at Sn sites, both the crystal structure and electronic band structure were facilitated, leading to the simultaneous enhancement of carrier mobility and effective mass. Resultantly, they obtained ultrahigh power factor PF of $\sim 85 \mu\text{W cm}^{-1} \text{K}^{-2}$ and ZT of ~ 1.4 at 300 K in the sample SnSe-9%Pb-1.2%Sr. On this basis, they fabricated thermoelectric devices to re-confirm the superior materials' performance, attaining rather high and stable conversion efficiencies approaching 9% at the temperature difference of only ~ 300 K. Both the materials' performance and the devices' efficiencies reported in this study showed obvious superior among the current low- to mid-temperature thermoelectrics. Moreover, the structural modulation strategy was well interpreted and verified by experimental observations as well as theoretical calculations. Therefore, I recommend this article be accepted for publication in Nature Communications. And before the final publication, there are some minor revisions to be resolved.

Response: We thank the reviewer for the recognition of our work and we have provided a point-to-point response as well as revised our manuscript as suggested.

Comment 1: I notice that this work can be regarded as a continuation of the paper published in 2021 (*Science* 2021, 373 (6554), 556-561), and the room-temperature ZT was further increased from ~ 1.2 to ~ 1.4 . While the power generation efficiency was largely enhanced, how about the thermoelectric cooling performance? Higher cooling temperature difference was also expected in this work.

Response: Thanks for your suggestion. We have further fabricated a 7-pair thermoelectric colling device using p-type SnSe-9%Pb-1.2%Sr and n-type commercial $\text{Bi}_2\text{Te}_{2.7}\text{Se}_{0.3}$ and evaluated its cooling performance with the commercial Z-Meters instrument. Results showed that the maximum cooling temperature difference ΔT_{max} reached ~ 50.4 K when the hot-end temperature was fixed at ~ 300 K, demonstrating significant enhancement compared with that of ~ 45.7 K in the published Science paper. The current results further advance SnSe crystals to be practically utilized in not only power generation, but also Peltier cooling. We have added the relevant discussions in the Revised Manuscript.

Revision:

Fig. 6d. The maximum cooling temperature difference ΔT_{\max} for the fabricated 7-pair thermoelectric devices using the SnSe-9%Pb-1.2%Sr crystals, and the ΔT_{\max} for the SnSe-9%Pb crystals was plotted for comparison.

Comment 2: The performance simulations from full-Pnma phase to full-Cmcm phase in Figure 1 are quite interesting. It is easy to understand that the two end curves in the 3D figures were obtained using the calculated electronic band structure of Pnma phase and Cmcm phase due to SR-XRD results. However, we can't obtain the 3D figures only from two end curves. How to obtain the simulated performance curves of the intermediate states between full-Pnma phase and full-Cmcm phase?

Response: Thanks for asking. As you said, it is easy to obtain the two end curves in Figure 1 by using the corresponding electronic band structures of full-Pnma and full-Cmcm phases from SR-XRD results. For the intermediate states between Pnma and Cmcm phases, we also calculated the performance parameters by using their corresponding electronic band structures. To establish the connection between the phase compositions and band structures, we consider the chosen angle 1 shown in Figure 4. First of all, the angles were determined to be $\sim 7.8^\circ$ and 0 for full-Pnma phase and full-Cmcm phase, respectively. Secondly, we chose an intermediate temperature, obtained its angle in the crystal structure, and thus determined the phase composition between full-Pnma phase and full-Cmcm phase with its corresponding band structure. For example, at 473 K, the angle was $\sim 7.3^\circ$ due to the refined crystal structure, corresponding to the intermediate state with $\sim 93.6\%$ Pnma or 6.4% Cmcm. Therefore, the electronic band structure for SnSe at 473 K was used to calculate the performance parameters of the intermediate state with 93.6% Pnma (also corresponding to 6.4% Cmcm). On this basis, several intermediate states between full-Pnma phase and full-Cmcm phase and their corresponding performance parameters were calculated, and we can finally obtain the simulated 3D curves shown in Fig. 1.

Comment 3: The bands merging temperature T_m for SnSe-9%Pb was announced to be

~ 423 K, not ~ 448 K. Please check it.

Response: Thanks for pointing out this question and we are sorry for our carelessness. We have fixed the relevant discussion and **Fig. 4** in the Revised Manuscript.

Comment 4: The multiband simulation curves of 3-band and 4-band models were plotted in Figure 2e-f. It is required to provide the simulation details in the Supplementary Information.

Response: Thanks for your suggestion and we have provided the multiband simulation details in the Revised Supporting Information.

Comment 5: In the first paragraph of the Introduction, κ_{lat} refers to the “lattice” part of κ_{tot} , not the “thermal” part.

Response: Sorry for this mistake and we have fixed it and carefully checked the whole manuscript for the similar questions.

Reviewer #3 (Remarks to the Author)

General Comment: Tin-selenide-based materials have been considered as one of the most promising thermoelectric materials and they have been widely investigated. In this manuscript, Zhao et al. reported a structural modulation strategy to improve the thermoelectric performances of p-type SnSe crystals. By alloying strontium at Sn sites, the crystal symmetry of low-temperature Pnma phase was modified, and the low- to mid-temperature thermoelectric performance were largely enhanced. Resultantly, an ultrahigh $ZT \sim 1.4$ at 300 K and $ZT_{\text{ave}} > 2.0$ among 300 – 773 K are obtained. The experimental and theoretical works in this manuscript are well performed, and the manuscript is well written. However, there are few points need to be addressed and clarified. I suggest the authors consider the follow comments before the manuscript can be accepted for publication

Response: We appreciate the reviewer 3 for the solid summary and good suggestions on our study. We hope that our point-to-point response and revisions will meet his/her demand and also improve the scientific quality of our manuscript.

Comment 1: SnSe possesses a layered structure and anisotropic thermoelectric properties, therefore, the thermal and electrical properties of the present sample should be measured in the same direction. The authors are suggested to provide the details on the measurement direction specifically in the manuscript.

Response: Thanks for your suggestion. In this study, we conducted the thermoelectric measurements all along the b -axis crystallographic direction in the cleavage plane of the crystals. The in-plane (b - c plane) orientations for crystal samples were determined using the X-ray back-reflection Laue measurement. And all samples for transport parameters measurements were cut and polished along the same b -axis direction. We have added the details in the Revised Manuscript and Supplementary Information.

Revision: The cleavage plane (400) of SnSe crystals was detected by X-ray back-reflection Laue to determine the in-plane (b - c plane) orientations. The crystals are placed according to their growth direction and their cleavage planes are facing to X-ray resource. The Laue patterns are obtained through collecting diffraction lines by a plate on a diffractometer operating at 20 kV and 18 mA. The thermoelectric transport parameters were all measured all along the b -axis direction.

Comment 2: As reported in the manuscript, the Na-doped SnSe-9%Pb- x %Sr ($x = 0.8, 1.2, 1.6, \text{ and } 2.0$) samples were prepared in this work. A high $ZT \sim 1.2$ at 300 K is obtained, which is much higher than that of the reported Pb-doped SnSe and the $\text{Bi}_{0.5}\text{Sb}_{1.5}\text{Te}_3$ materials. However, the role of Pb alloying is not discussed in the manuscript. The authors should address this issue and make the comparison with reported Pb-doped SnSe sample.

Response: Thanks for your suggestion. In this study, we realized the structural modulation by alloying Sr in Pb-alloyed SnSe. The Pb-alloyed SnSe crystals were systematically investigated in our previous work (*Science* 2021, 373 (6554), 556-561). Briefly speaking, in Pb-alloyed SnSe, we for the first time revealed the significantly

promoted multiband synglisis including the momentum alignment of bands merging process from VBM (1 and 2) merging into VBM (1+2), and the energy alignment of bands aligning between VBM 3 and VBM (1+2). The momentum alignment from two bands (before merging) to one band (after merging) largely increased the carrier mobility, while the energy alignment of multiple bands aligning substantially strengthened effective mass. Therefore, the promoted multiband synglisis by Pb alloying in SnSe realized the significant synergy between the competing parameters carrier mobility and effective, leading to ultrahigh power factor and ZT values near room temperature and over the whole temperature range. On the basis of Pb alloying, we introduced Sr alloying in SnSe in this study, realized the structural modulation of not only modified electronic band structure but also crystal structure, and thus further optimized the thermoelectric performance and efficiency in p-type SnSe crystals. To better demonstrate the comparison before and after Sr alloying, we have also plotted the performance curves as well as the characteristic temperatures for Pb-alloyed SnSe over the whole Revised Manuscript.

Comment 3: As shown in Fig. 5, both the *Pnma* phase and the high-symmetry phase are observed in the as-prepared SnSe-9%Pb-1.2%Sr sample. Thus, the two phases should be considered during Rietveld refinements of p-type SnSe-9%Pb-1.2%Sr sample. Moreover, the refinements results for each sample should be provided in form of table in Supplementary Information.

Response: Thanks for your suggestion. During the Rietveld refinements, we used GSAS and Highscore software to refine XRD patterns. To determine whether the parameter adjustment is appropriate during the refinement process, we use the χ^2 to reflect the quality of the simulations. The smaller χ^2 we have, the smaller deviation we can obtain in the accuracy of the Rietveld refinement. We have also attempted to consider both phases during the refinements. Taking the data at ~ 300 K as examples for comparison, we plotted the patterns and refinement results considering only *Pnma* phase and both phases, as shown in Fig. R1. It can be clearly seen that Fig. R1a depicts better accuracy than Fig. R4b, and the χ^2 values are 0.9679 and 1.757, respectively.

Fig. R1. SR-XRD patterns and Rietveld refinements for p-type SnSe-9%Pb-1.2%Sr at 300 K considering (a) *Pnma* phase only and (b) both phases.

This might be because that the detection limit of SR-XRD restrict the experimental

observation of the nanoscale second phases. Besides, the diffraction patterns of *Pnma* and *Cmcm* phases are extremely close to each other, further limiting the detection of SR-XRD. Therefore, even we observed the *Pnma* and *Cmcm* phases of SnSe at the microscale, to obtain a better refinement accuracy, we provided the Rietveld refinements considering the single phase and by identifying the (011) characteristic peak at around $2\theta = 13.0^\circ$, we can easily distinguish between the *Pnma* and *Cmcm* phases. Moreover, as you suggested, we have provided the refinement results consisting the lattice parameters and atomic positions in **Supplementary Table 1, 2** in the Revised Supplementary Information.

Comment 4: In Fig. 5h, Fig. 5i and Fig. 5j, the black backgrounds of text box in the middle of the figures affected the display of pictures.

Response: Thanks for your suggestion. We have deleted the mentioned black backgrounds in **Fig. 5**.

Revision:

Fig. 5. Microstructure characterization on the p-type SnSe-9%Pb-1.2%Sr crystal.

Comment 5: In addition, it is suggested to polish the English of the whole manuscript.

Response: Thanks for your suggestion. We have checked the manuscript carefully and made the relevant corrections.

REVIEWERS' COMMENTS

Reviewer #1 (Remarks to the Author):

I have no other questions.

Reviewer #2 (Remarks to the Author):

Thanks for resubmitting your paper. After reviewing it, I believe the answers could address my questions, and the author provides a comprehensive response to each point.

I am satisfied with the revisions and agree that the manuscript is ok for publication. I appreciate your commitment and hard work, and I look forward to seeing the final version.

Reviewer #3 (Remarks to the Author):

The authors have addressed all my concerns and made the necessary changes to the manuscript. I recommend it for publication in Nature Communications.

Authors Response

Reviewer #1 (Remarks to the Author)

I have no other questions.

Response: We thank Reviewer 1 for the positive comments and general suggestions on our study. We also appreciate those critical suggestions which help further improve the scientific quality of our manuscript.

Reviewer #2 (Remarks to the Author)

Thanks for resubmitting your paper. After reviewing it, I believe the answers could address my questions, and the author provides a comprehensive response to each point. I am satisfied with the revisions and agree that the manuscript is ok for publication. I appreciate your commitment and hard work, and I look forward to seeing the final version.

Response: We thank Reviewer 2 for the recognition of our work and the suggestions have undoubtedly improved our manuscript.

Reviewer #3 (Remarks to the Author)

The authors have addressed all my concerns and made the necessary changes to the manuscript. I recommend it for publication in Nature Communications.

Response: We appreciate Reviewer 3 for the solid summary and good suggestions on our study. We believe that we have improved the scientific quality of our manuscript as suggested.